# Prospective Validation and Refinement of a Population Pharmacokinetic Model of Fludarabine in Children and Young Adults Undergoing Hematopoietic Cell Transplantation

**DOI:** 10.3390/pharmaceutics14112462

**Published:** 2022-11-15

**Authors:** Jordan T. Brooks, Belen P. Solans, Ying Lu, Sandhya Kharbanda, Christopher C. Dvorak, Nahal Lalefar, Susie Long, Ashish O. Gupta, Biljana Horn, Jatinder K. Lamba, Liusheng Huang, Beth Apsel-Winger, Ron J. Keizer, Rada Savic, Janel Long-Boyle

**Affiliations:** 1Department of Clinical Pharmacy, University of California San Francisco, San Francisco, CA 94143, USA; 2Department of Bioengineering and Therapeutics, University of California San Francisco, San Francisco, CA 94143, USA; 3Department of Pediatrics, University of California San Francisco, San Francisco, CA 94143, USA; 4Benioff Children’s Hospital of Oakland, University of California San Francisco, Oakland, CA 94609, USA; 5Department of Pediatrics, University of Minnesota Masonic Children’s Hospital, Minneapolis, MN 55454, USA; 6Department of Pharmacotherapy and Translational Research, College of Pharmacy, University of Florida, Gainesville, FL 32610, USA; 7Department of Pediatrics, University of Florida, Gainesville, FL 32603, USA; 8Insight RX, San Francisco, CA 94104, USA

**Keywords:** population pharmacokinetics, pediatric pharmacology, hematopoietic cell transplant conditioning

## Abstract

Fludarabine is a nucleoside analog with antileukemic and immunosuppressive activity commonly used in allogeneic hematopoietic cell transplantation (HCT). Several fludarabine population pharmacokinetic (popPK) and pharmacodynamic models have been published enabling the movement towards precision dosing of fludarabine in pediatric HCT; however, developed models have not been validated in a prospective cohort of patients. In this multicenter pharmacokinetic study, fludarabine plasma concentrations were collected via a sparse-sampling strategy. A fludarabine popPK model was evaluated and refined using standard nonlinear mixed effects modelling techniques. The previously described fludarabine popPK model well-predicted the prospective fludarabine plasma concentrations. Individuals who received model-based dosing (MBD) of fludarabine achieved significantly more precise overall exposure of fludarabine. The fludarabine popPK model was further improved by both the inclusion of fat-free mass instead of total body weight and a maturation function on fludarabine clearance. The refined popPK model is expected to improve dosing recommendations for children younger than 2 years and patients with higher body mass index. Given the consistency of fludarabine clearance and exposure across its multiple days of administration, therapeutic drug monitoring is not likely to improve targeted exposure attainment.

## 1. Introduction

Allogeneic hematopoietic cell transplantation (HCT) is used to treat many malignant and non-malignant disorders in children. Preparative drug regimens prior to HCT commonly utilize the nucleoside analogue fludarabine, which possesses both antileukemic activity and strong immunosuppressive activity that potentially enhances stem cell engraftment. After fludarabine phosphate is infused, it is rapidly dephosphorylated to form the principle circulating metabolite f-ara-a, which then enters the cells via nucleoside membrane transporters and is eventually phosphorylated into the cytotoxic, active form f-ara-ATP. While drug toxicity is less concerning with f-ara-a when compared with busulfan, cyclophosphamide and other myeloablative alkylating agents, f-ara-a exposure variability is still potentially linked with poor transplant outcomes including nonrelapse mortality and impaired immune reconstitution. Additionally, fludarabine neurotoxicity is a rare but serious complication of fludarabine which is thought to be dose-dependent [1].

Our understanding of f-ara-a pharmacokinetics (PK) in pediatrics has been based on limited data generated from two clinical trials including a limited sample size, excluding children <1 year of age, and treating patients concomitantly with cytarabine in the context of relapsed leukemias [2,3]. Within the adult population, studies of f-ara-a PK in HCT populations found that patient specific covariates, including body weight and renal function (two covariates of great variability within the pediatric population), impact estimations of f-ara-a clearance (CL) [4,5,6]. Recent studies have corroborated that body weight and renal function impact f-ara-a exposure within the pediatric population [7,8]. Both studies developed population PK models from which individualized f-ara-a doses can be generated for pediatric patients [7,8]. Additionally, in the Ivaturi model characterized both f-ara-a as well as it intracellular, active f-ara-ATP concentrations in peripheral blood mononuclear cells. They found that with each additional consecutive daily dose of fludarabine, intracellular concentrations of f-ara-ATP decrease while circulating f-ara-a concentrations remain consistent, suggesting a degree of saturation in the transport of f-ara-a intracellularly [8].

Beyond population PK model development, evaluation of optimal f-ara-a exposure has also found early agreement within the literature. In a previous study published by our group, pediatric subjects with f-ara-a cumulative area-under-the-curve (cAUC) above 15 mg*h/L and below 19 mg*h/L had the highest probability of a positive outcome for patients transplanted for malignancy, while no such trend was readily apparent within the limited set of subjects transplanted for nonmalignant conditions [8]. Though PK parameters were well estimated and goodness of fit (GOF) plots were acceptable, high residual unexplained variability remained within this model [8]. In a study including pediatric and adult subjects, optimal f-ara-a exposure was found to be 20 mg*h/L with those who experienced supratherapeutic exposures having increased risk of impaired immune reconstitution and those who experienced subtherapeutic exposures having increased risk of nonrelapse mortality and graft failure [9]. Improved event-free survival was found at the optimal f-ara-a exposure for both malignant and non-malignant disorders [9].

The aim of the current study was to validate and refine the pediatric population PK model, previously developed by our group utilizing sparse PK sampling in a prospective multicenter pediatric clinical trial [8]. The second aim was to compare f-ara-a exposure variability between patients dosed based on model informed dosing and those receiving fludarabine dosing based on weight or body surface area (BSA).

## 2. Materials and Methods

Patients recruited to participate in the study met protocol-specific eligibility criteria for transplantation, planned to undergo an allogeneic or autologous HCT that included intravenous fludarabine phosphate as part of the conditioning regimen, and included pediatric (0–17 years old) or young adult (18–22 years old). Participating HCT centers included the University of California, San Francisco (UCSF), UCSF Benioff Children’s Hospital of Oakland, University of Minnesota, and Rady Children’s Hospital San Diego were recruited between October 2019 and February 2022. F-ara-a dosing strategies employed were institution-specific; patient enrollment in this study did not change the clinical course of HCT. UCSF was the only center to utilize the Ivaturi model for Model Based Dosing (MBD) of f-ara-a via a Bayesian dosing estimation platform (InsightRX Nova) [8]. Patient renal function was assessed by estimation of creatinine clearance (CrCL) via the Bedside Schwartz equation for patients under 17 years old and the Cockcroft-Gault equation for those over 18 years old, and CrCL estimates were capped at 150 mL/min/1.73 m^2^ prior to use within the popPK model. Pediatric fat-free mass was estimated for patients over 3 years old and under 18 years old using the method published by Al-Sallami et al. [10].

For MBD, the clinical team used patient-specific information and cAUC targets ranging from 16–20 mg*h/L to generate the f-ara-a dosing regimen for each patient. Dosing strategies employed by contributing sites, and for a minority of patients from UCSF, included BSA-based dosing (mg/m^2^) for patients greater than 10 kg or 2 years of age and weight-based dosing (mg/kg) for individuals less than 10 kg or 2 years of age. F-ara-a doses were given every 24 h from three to five days, depending on the patient’s conditioning regimen.

Given the rich sampling strategy utilized to build the original popPK model, plasma f-ara-a concentrations were collected using a sparse sampling strategy, with two measurements drawn per patient, to minimize extraneous blood draws for patients during HCT. The first f-ara-a concentration was drawn 0–2 h after the end of infusion, and the second f-ara-a concentration was drawn any time 2–24 h after the end of infusion. Draws were timed to optimize clinical care of the patients during HCT, and specific times of collection were recorded for each patient draw. Patient characteristics were collected at the time of transplant that included age, gender, ethnicity, total body weight, height, and markers of renal function. Transplant-specific characteristics included diagnosis, conditioning regimen, stem cell source, and degree of HLA mismatch. Patient information was uploaded by HCT center research coordinators and stored in a secure REDCap database. Plasma samples were analyzed for f-ara-a using a previously validated liquid chromatography-tandem mass spectrometry method [11]. The F-ara-A assay was linear in the range of 2 ng/mL to 800 ng/mL. During the sample analysis, the mean accuracies (mean ± coefficient of variation) of the f-ara-a assay were 100.7% ± 8.3, 103.7% ± 8.6, and 101.2% ± 11.8 at low-, medium-, and high-quality control levels, respectively.

Software utilized to validate and refine the previously published model included NONMEM v7.5 (ICON), Perl Speaks NONMEM v4.8.1, and PiranaJS (beta version). For model refinement, the prospective data set was added to the original data set to make a combined data set. Determination of the model with the best fitness was based on –2 × log likelihood which is presented by NONMEM as the objective function value (OFV). When comparing models, a difference in the OFV of (dOFV) of 3.84, 7.88, and 10.83 are considered significant at the 0.05, 0.005, and 0.001 levels, respectively. Beyond dOFV comparisons, goodness-of-fit (GOF) plots and prediction corrected visual predictive checks (VPC) were utilized to assess model fitness. The precision of the final refined model parameters was evaluated by performing a 1000 non-parametric bootstrap analysis and reporting the 95% confidence intervals (CI) for each parameter.

Simulations were run to compare the original and refined models’ accuracy in attaining a cAUC target of 20 mg·h/L in individuals aged less than 2 years old and individuals with a body mass index (BMI) greater than 30 kg/m^2^. The simulated data sets included 1000 individuals, assigning height and weight according to age and sex as described in growth curve distributions listed by the Centers for Disease Control [12]. For both the refined and original model, 4 doses were assigned according to the typical model estimate of clearance, and f-ara-a cAUC was then predicted by NONMEM utilizing the refined model.

Variance in model-generated predicted cAUCs were statistically compared between patients who received a traditional dosing strategy and those who received MBD utilizing Levene’s test. Additionally, a descriptive evaluation of model accuracy was made. The current prospective data set was combined with the original model building data set prior to model refinement.

## 3. Results

Of those eligible, a total of 53 patients consented to participate in this study for model validation, with subject demographics summarized in Table 1. This sample of subjects had median age of 8.0 years with seven individuals ≤2 years old (13%), and a median weight of 23.6 kg. Of the 53 subjects, 24 received MBD while 29 received traditional weight or BSA based dosing of f-ara-a.

A total of 101 f-ara-a concentrations were quantifiable and utilized in model validation and refinement. A prediction corrected VPC of concentrations over time after dose suggests the popPK model well-predicted the concentrations observed in this prospective data set (Figure 1). The total percent error of model-predicted concentrations was 6.72% as calculated by Equation (1).
(1)$$PE=\frac{100}{n}\sum_{n}^{i}\left(\frac{\left[Prediction\right]-\left[Observation\right]}{\left[Observation\right]}\right)$$

For the 24 subjects who received MBD targeting a specific cAUC goal, model accuracy was evaluated as the percentage difference of predicted cAUC to goal cAUC. The model’s median accuracy was 94% (range: 67–118%).

A comparison of variance between f-ara-a cAUCs between those who received MBD versus those who received traditional f-ara-a dosing via Levene’s test found the two groups had statistically different variance (*p* < 0.005). MBD resulted in more precise attainment of f-ara-a cAUC when compared with traditional dosing strategies (Figure 2).

Since the current data set only contains sparse f-ara-a concentrations and no f-ara-ATP concentrations, the parameters of intracellular compartment described in the original model were fixed and not further estimated. Addition of a maturation function, as defined by Equation (2),
(2)$$F_{mat}=1-e^{\left(-Age\;*\;\theta \right)}$$
where θ is an estimated parameter that impact the shape of the exponential curve, on CL was found to significantly improve the model (dOFV-12.3). Additionally, replacing weight with fat-free mass significantly improved model fit (dOFV-11). Accounting for participant ancestry within the model did not improve the model fit. The refined model for f-ara-a CL is presented in Equation (3):(3)$$CL_{individual}=CL_{population}\times F_{mat}\times {\left(\frac{FFM}{12\;kg}\right)}^{0.75}\times \left(1+\left(CrCL-100\;\mathrm{mL}/\min/1.73m^{2}\right)\right)\times CrCL_{eff}$$

Given the primary renal clearance of f-ara-a, both the factors accounting for CrCL and maturation function in the overall f-ara-a CL are reflective of renal function; however, the maturation function is thought to account for development related factors impacting f-ara-a clearance not detectable by the approximation of CrCL as seen in Figure 3. Final refined model parameter estimates are included in Table 2 and a prediction corrected visual predictive check of the refined model is included in Appendix A.

The simulated data suggest that the refined model doses improved the likelihood for achieving a specific exposure for both individuals less than 2 years old and for those with BMI greater than 30 kg/m^2^ (Figure 4). For individuals less than 2 years old, doses assigned by the refined model resulted in a predicted cAUC median (range) of 20.3 (17.1–32.0), whereas doses assigned by the original model resulted in a predicted cAUC median (range) of 25.0 (20.3–33.0). For individuals with a BMI greater than 30 kg/m^2^, doses assigned by the refined model resulted in a predicted cAUC median (range) of 20.9 (18.3–21.4), whereas doses assigned by the original model resulted in a predicted cAUC median (range) of 30.6 (27.1–42.2).

## 4. Discussion

As popPK models are developed and translated into clinical practice, it is imperative that they are both prospectively validated with new data not utilized to develop the model and refined with the increased sample size that comes with the validation process. This is especially important when implementing popPK-based, precision dosing within vulnerable populations which generally have sample size limitations and large pharmacologic variability.

In terms of validation, the previously published f-ara-a popPK model captures the prospectively observed f-ara-a concentrations well, which is particularly notable given the lack of therapeutic drug monitoring available to adjust between doses. Unlike other medications often used in conjunction with fludarabine such as busulfan, the patient-specific PK parameter estimates appear to stay quite consistent over the course of f-ara-a administration. While busulfan is metabolized by the liver and its clearance is susceptible to drug–drug interactions and its own hepatotoxic effects, fludarabine is renally eliminated and its clearance appears consistent across its course of administration. Based on the evidence present in this study, there is likely minimal benefit to recommending therapeutic drug monitoring for f-ara-a at this time, as target exposures were easily attained with the initial dose estimate suggested by the model. Compared with traditional dosing strategies, utilizing MBD via the described model improves the precision of f-ara-a cAUC attainment. In particular, MBD avoids the extreme cAUC lows and highs previously associated with worse clinical outcomes [9].

While the original popPK model performed well with patients greater tha 2 years of age and with normal BMIs, the additional f-ara-a PK data measured in this study particularly enabled us to study differences within f-ara-a PK within children under 2 years of age and patients with higher BMIs. The expansion of participants less than 2 years of age enabled us to approximate a maturation function on f-ara-a CL that describes how CL increases particularly over the first year of life. Caution should be used in applying the developed maturation function in individuals younger than 5 months of age given this is the model has not been validated for such individuals and, therefore, the model could suggest an erroneously low dose of fludarabine. While not a mechanistic description, the maturation function is in alignment with f-ara-a being primarily cleared renally and that kidney maturation of infants typically reaches adult equivalence by ~1–2 years of age [13]. Further work to characterize the specific mechanisms of renal CL including the contributions of glomerular filtration and active tubular secretion is warranted. Beyond characterizing a maturation function on CL, replacing weight with fat-free mass on estimates of CL, Vc, Intercompartmental CL, and Vp additionally improved the model. This suggests f-ara-a does not likely partition into adipose tissue and, therefore, excess mass contributed by fat minimally impacts f-ara-a exposure and distribution in the body. Adopting fat-free mass into the model will enable us to improve dose recommendations for high BMI individuals, preventing doses which are erroneously and disproportionally too high.

Given improvement in model fit after including maturation function and fat-free mass as well as re-estimating the PK parameters with the expanded data set, the model accuracy is expected to improve in use within future patients. While patient-reported ancestry information was initially explored, it was not incorporated in the final refined model at this time. Furthermore, inclusion of ancestry factors needs to be completed in an equitable manner, supported by a more robust pharmacogenetic analysis.

## 5. Conclusions

This study indicates the previously developed model by Ivaturi et al. well describes f-ara-a PK in the pediatric population undergoing hematopoietic stem cell transplantation. Pediatric patients who receive MBD of f-ara-a achieve more precise cAUCs preventing extreme low or high f-ara-a exposure which are more likely to cause adverse events. Including a maturation function on the individual estimate of CL and using fat-free mass instead of weight for estimates of CL and volume of distribution improved the model fitness, resulting in a more robust accounting for both children less than 2 years old and overweight individuals. Since this model enables the achievement of a targeted exposure over a set period, there is no longer a need to give additional doses beyond 4 days of fludarabine and suggesting that 5 days of fludarabine, which is still a current practice in some centers, is not beneficial as active, intracellular concentrations f-araATP decrease with extending fludarabine therapy by an additional day. Given the consistency of f-ara-a CL over multiple doses, it is unlikely that therapeutic drug monitoring would improve cAUC target attainment.

## Figures and Tables

**Figure 1 pharmaceutics-14-02462-f001:**
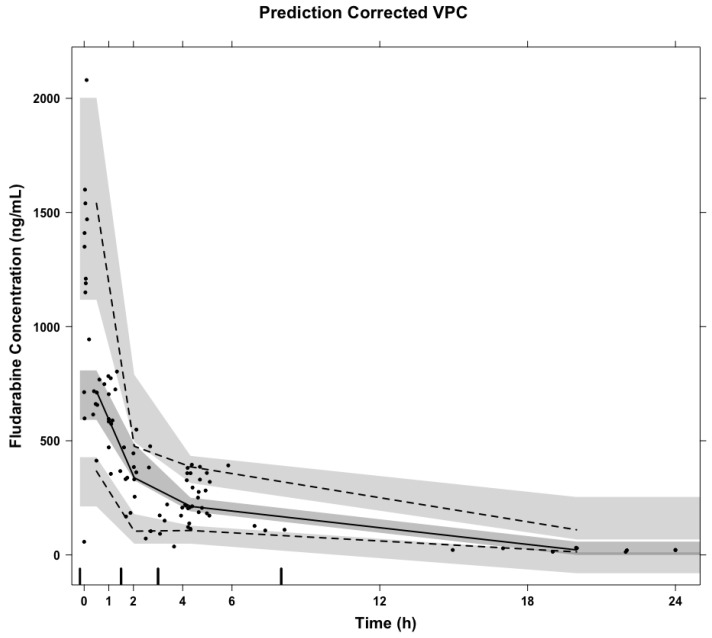
Prediction corrected visual predictive check of the original popPK f-ara-a concentrations (ng/mL) over time after dose (h) of the validation cohort. The solid and dashed lines represent the median, 5th, and 95th percentile of observed f-ara-a concentrations, respectively. The shaded areas represent the prediction intervals generated by the popPK model for the median, 5th, and 95th percentiles, respectively.

**Figure 2 pharmaceutics-14-02462-f002:**
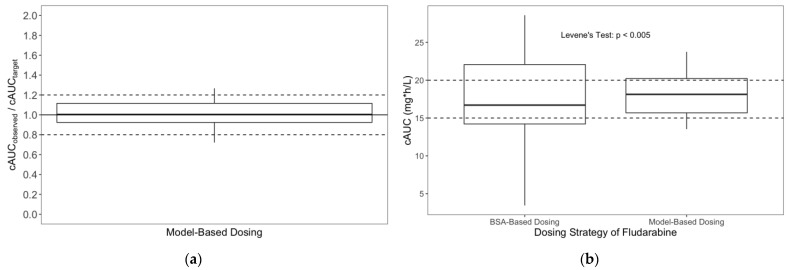
(**a**) Model performance presented as the ratio of cAUC_observed_/cAUC_target_ in 24 subjects who received MBD of f-ara-a prior to HCT. (**b**) A comparison of the variability in cAUC between subjects undergoing BSA-based dosing and pre-emptive MBD of f-ara-a prior to HCT.

**Figure 3 pharmaceutics-14-02462-f003:**
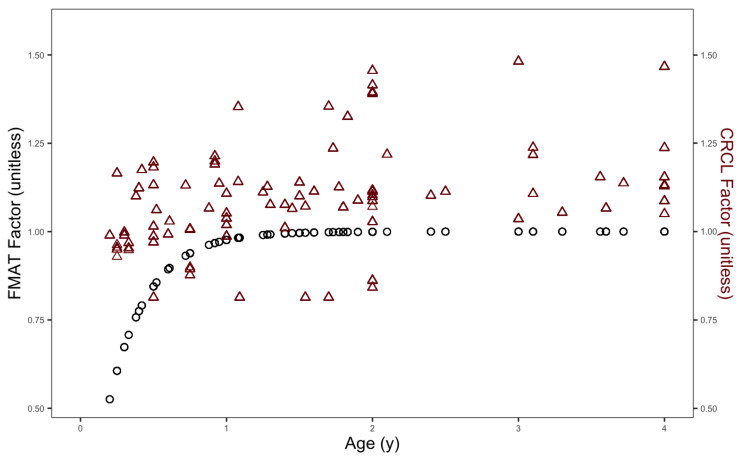
Maturation function factor on f-ara-a CL (left axis, circles) and CrCL factor on f-ara-a CL (right axis, triangles) over age in years of the combined data set used to refine the model. CrCL factor does account for age-related maturation of CL of f-ara-a.

**Figure 4 pharmaceutics-14-02462-f004:**
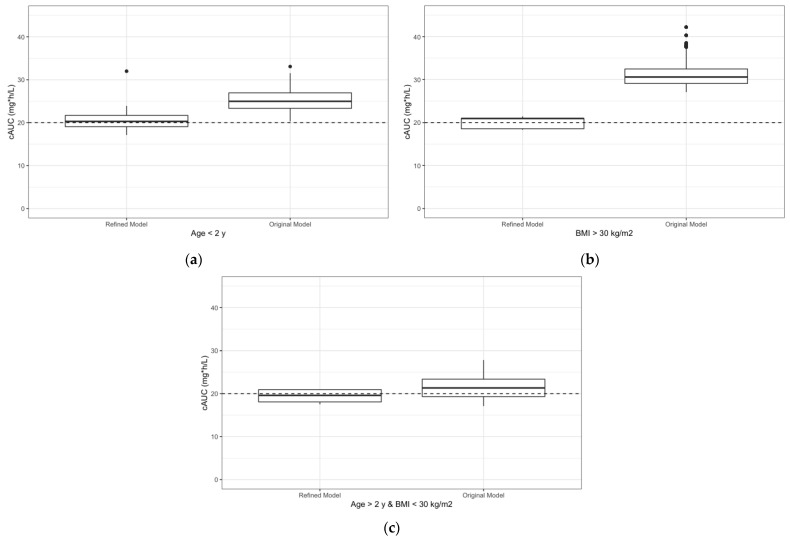
Simulated cAUC values to compare doses assigned by the refined model and the original model. (**a**) Predicted cAUCs of individuals less than 2 years old separated by which model was utilized to assign doses. (**b**) Predicted cAUCs of individuals with BMI greater than 30 kg/m^2^ separated by model was utilized to assign doses. (**c**) Predicted cAUCs of individuals greater than 2 years old and BMI less than 30 kg/m^2^ separated by model was utilized to assign doses.

**Table 1 pharmaceutics-14-02462-t001:** Subject Demographics and Baseline Characteristics.

New Subjects for Model Validation		Combined New and Original Subjects for Model Refinement	
Characteristic	Value	Characteristic	Value
Participants—n	53	Participants—n	186
Dosing Strategy—n (%)		Dosing Strategy—n (%)	
Traditional	29 (55%)	Traditional	162 (87%)
40 mg/m^2^	20	40 mg/m^2^	75
12.5–30 mg/m^2^	6	12.5–30 mg/m^2^	46
0.9–1.33 mg/kg	3	0.9–1.33 mg/kg	41
Model-based dosing	24 (45%)	Model-based dosing	24 (13%)
Sex—n (%)		Sex—n (%)	
Females	19 (36%)	Females	80 (43%)
Males	34 (64%)	Males	106 (57%)
Weight, median (range), kg	23.6 (7.3–135)	Weight, median (range), kg	21.6 (3–135)
Height, median (range), cm	121.7 (67–186.1)	Height, median (range), cm	112.4 (50–186.7)
Age, median (range), year	8.0 (0.4–22)(7 subjects ≤ 2 year)	Age, median (range), year	6.4 (0.2–22)(55 subjects ≤ 2 year)
CrCL, median (range), ml/min/1.73 m^2^	137 (53–337)	CrCL, median (range), mL/min/1.73 m^2^	149 (120–193)

**Table 2 pharmaceutics-14-02462-t002:** Final PopPK Model Parameter Estimates.

	Final Model		Bootstrap	
Parameter	Value	RSE *	Value	95% CI
Effect of CrCL f-ara-a CL	0.00186	32.8	0.0018	0.001–0.004
Vc (L/12 kg of FFM)	11.5	4.9	11.4	10.22–12.57
Intercompartmental Clearance (L/h/12 kg of FFM)	1.47	9.3	1.48	1.23–1.79
Vp (L/12 kg of FFM)	8.48	4.6	8.53	7.75–9.29
Maturation Function Shape	3.54	13.8	3.57	2.65–4.75
f-ara-a additive residual unexplained variability	0.36	12.6	0.36	0.27–0.45
Interindividual Variability on CL ^†^	0.117	20.7	0.118	0.07–0.26
Interindividual Variability on Vc ^†^	0.303	18.1	0.314	0.21–0.43
Interindividual Variability Correlation CL-Vc	0.141		0.149	0.09–0.21

RSE indicates relative standard error; Vc volume of the central compartment; Vp, volume of the peripheral compartment. * Relative standard error expressed as % mean. ^†^ Presented as %CV.

## Data Availability

Not applicable.

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
