# Peer review of "Prospective Validation and Refinement of a Population Pharmacokinetic Model of Fludarabine in Children and Young Adults Undergoing Hematopoietic Cell Transplantation"

_pharmaceutics, 2022, doi:10.3390/pharmaceutics14112462_

Round 1

Reviewer 1 Report

The text is very simple. The method, results and discussion need the several changes. There is few references

Line  84 -  cells HCT – what  is the origin?

                    Age children and Young adult?

Line 142 – the text  A total of 53 subjects all of whom received an allogeneic HCT completed PK analysis 140 and were used to assess model validity, with subject demographics summarized in Table 141 1. This sample of subjects had median age of 8.0 years with seven individuals ≤ 2 years 142 old (13%), and a median weight of 23.6 kg. Of the 53 subjects, 24 received MBD while 29 143 received traditional weight or BSA based dosing of f-ara-a.. this is methodology

Tab 1 is information about methodology of the experiment. This table must be in the methods.

Line 211 - In terms of validation, the previously published f-ara-a popPK model captures the  prospectively observed f-ara-a concentrations well, which is particularly notable given the lack of therapeutic drug monitoring available to adjust between doses. Unlike other medications often used in conjunction with fludarabine such as busulfan, the patient-specific  PK parameter estimates appear to stay quite consistent over the course of f-ara-a administration. Based on the evidence present in this study, there is likely minimal benefit to recommending therapeutic drug monitoring for f-ara-a at this time. Compared with traditional dosing strategies, utilizing MBD via the described model improves the precision of f-ara-a cAUC attainment. In particular, MBD avoids the extreme cAUC lows and highs  previously associated with worse clinical outcomes [9]. It is necessary a better discussion.

Line 222 - under 2 years of age. And the others children?

line 230 -  Further work to characterize the specific mechanisms of renal CL including the contributions glomerular filtration and active tubular secretion is warranted. Beyond characterizing a maturation  function on CL, replacing weight with fat-free mass additionally improved the model,  suggesting f-ara-a does not likely partition into adipose tissue and enabling us to improve  dose recommendations for high BMI individuals.  It is necessary a better discussion.

Line 246 – this  line has a reference. Conclusions dont have reference. It is better remove the reference

Author Response

"The text is very simple. The method, results and discussion need the several changes. There is few references"

Response:

Historically, the implementation of population pharmacokinetic models has been hampered by several logistical issues associated with the science, including unnecessary or overly complicated manuscripts of which a PK-novice or clinician cannot understand.  Given the know association between optimal fludarabine exposure and reduction in disease relapse, there is significant interest within both the HCT and CAR-T community towards international adoption of model-based dosing.  Thus, we purposely and artfully have written this manuscript in a way which ensures reproducibility of the validation process for trained pharmacometricians with significant knowledge of the statistical methods and a novice clinician looking to implement this science into the clinic with relative ease.  We feel strongly this is important for readers of various levels of understanding and clinical expertise.

"Line  84 -  cells HCT – what  is the origin?"

Response:

HCT-  hematopoietic cell transplantation is a standard abbreviation used for hematopoietic stem cells obtained from the three standard sources:  bone marrow, peripheral blood mononuclear cells, and banked umbilical cord blood.

"Age children and Young adult?"

Response: 

As noted in the methods and results (Table 1) the patient population included in this analysis includes children and young adults, ages 0-22 years of age.  This is the standard range used clinical for pediatric indications for stem cell transplantation.  The text has been clarified to reflect this analysis includes both children and young adults.

"Line 142 – the text  A total of 53 subjects all of whom received an allogeneic HCT completed PK analysis 140 and were used to assess model validity, with subject demographics summarized in Table 141 1. This sample of subjects had median age of 8.0 years with seven individuals ≤ 2 years 142 old (13%), and a median weight of 23.6 kg. Of the 53 subjects, 24 received MBD while 29 143 received traditional weight or BSA based dosing of f-ara-a.. this is methodology"

Response:

We respectfully disagree and wish to keep this content in the results.  The eligibility/recruitment criteria is provided in the methods sections, and explains who was eligible for inclusion in the analysis.  However, not all subjects that met eligibility criteria are included in the final dataset used for analysis.  Text has been added to ensure clarity between eligibility and inclusion in the final dataset.

"Tab 1 is information about methodology of the experiment. This table must be in the methods."

Response

As previously discussed, we have included additional text to ensure clarification between recruitment/eligibility and those subjects included in the final analysis.

"Line 211 - In terms of validation, the previously published f-ara-a popPK model captures the  prospectively observed f-ara-a concentrations well, which is particularly notable given the lack of therapeutic drug monitoring available to adjust between doses. Unlike other medications often used in conjunction with fludarabine such as busulfan, the patient-specific  PK parameter estimates appear to stay quite consistent over the course of f-ara-a administration. Based on the evidence present in this study, there is likely minimal benefit to recommending therapeutic drug monitoring for f-ara-a at this time. Compared with traditional dosing strategies, utilizing MBD via the described model improves the precision of f-ara-a cAUC attainment. In particular, MBD avoids the extreme cAUC lows and highs  previously associated with worse clinical outcomes [9]. It is necessary a better discussion."

Response

We have clarified/edited the text explaining why the addition of therapeutic drug monitoring is unlikely much more added benefit to model-based dosing for several reasons including clearance/dose interval (e.g. lack of inter-occasion variability), fewer drug interactions, and renal elimination (vs hepatic).

"Line 222 - under 2 years of age. And the others children?"

Response: 

We have clarified the x-axis for Figure 4C and now include box plots for the following groups:

Less than 2 years of age

Greater than 2 years of age and body mass index <30

Body mass index greater than 30

"line 230 -  Further work to characterize the specific mechanisms of renal CL including the contributions of glomerular filtration and active tubular secretion is warranted. Beyond characterizing a maturation function on CL, replacing weight with fat-free mass additionally improved the model,  suggesting f-ara-a does not likely partition into adipose tissue and enabling us to improve  dose recommendations for high BMI individuals.  It is necessary a better discussion."

Response: 

We have added additional text to expand our discussion on the distribution of fludarabine and the limited clearance that occurs in adipose tissue.

"Line 246 – this  line has a reference. Conclusions dont have reference. It is better remove the reference"

Response: 

We have removed citations in the conclusion section of the manuscript.

Reviewer 2 Report

1.       The author must differentiate the term pediatric and young adults.

2.       Elaborate the sparse sampling technique in the methodology section.

3.       Add more statistical tools in the manuscript.

Author Response

“1.     The author must differentiate the term pediatric and young adults.”

We have altered the verbiage and updated our defined age ranges for these terms throughout the text.

“2.       Elaborate the sparse sampling technique in the methodology section.”

We have added further information to the methodology section to clarify what we mean by sparse sampling.

“3.       Add more statistical tools in the manuscript.”

As we have followed standard non-linear mixed effects modeling processes, which are statistically robust and have allowed us to answer the clinical research questions driving the project with confidence and certainty, we do not feel additional statistical tools are needed to support the findings of this paper. 

Reviewer 3 Report

The authors used nonlinear mixed-effects modelling to develop a fludarabine popPK model and the fludarabine popPK model was further improved by both the inclusion of fat-free mass instead of total body weight and a maturation function on fludarabine clearance. The method is somewhat unclear and it is difficult to understand how PK parameters were derived and their biological appropriateness of them. As written, it is unclear if estimated values are biologically plausible. The authors should build the model from a biological point of view, and explain the model with this mindset so that the readers can easily understand how the developed model addressed the aim of the study. The model should be included in this manuscript. 

Author Response

"The authors used nonlinear mixed-effects modelling to develop a fludarabine popPK model and the fludarabine popPK model was further improved by both the inclusion of fat-free mass instead of total body weight and a maturation function on fludarabine clearance. The method is somewhat unclear and it is difficult to understand how PK parameters were derived and their biological appropriateness of them. As written, it is unclear if estimated values are biologically plausible. The authors should build the model from a biological point of view, and explain the model with this mindset so that the readers can easily understand how the developed model addressed the aim of the study. The model should be included in this manuscript."

Response:

The detailed methods for model development have been rigorously peer-reviewed and previously published (Ivaturi et al. 2017, BBMT).  Briefly, as part of the standard model-building process and nonlinear mixed effects methods, the assessment for the physiologic plausibility for both the base model and clinical covariates were evaluated by the appropriate statistical tests. The final model consisted of a 2-compartment model with linear elimination with actual body weight and creatinine clearance as independent predictors of drug clearance.

The current manuscript describes assessment and further refinement of the published model with the use of an independent external dataset.  The reassessment of model-performance after clinical implementation is a critical part of the ongoing validation process:  As more data becomes available (e..g. more patients of variable age and clinical factors) refinement of the model estimated parameters can be performed or additional covariates can be tested for an effect on drug clearance and exposure.

The final model of the clearance parameter utilized to guide dosing for cAUC target attainment is provided in the current version of the manuscript (equation 3).  We have added some additional text to ensure clarity of the model validation purpose and process.

Round 2

Reviewer 1 Report

the text is good now